# Silver Nanoparticles (Ag NPs) Boost Mitigation Powers of *Chenopodium Quinoa* (Q6 Line) Grown under In Vitro Salt-Stressing Conditions

**Rida Shibli [1,2,*], Ruba Mohusaien [2], Rund Abu-Zurayk [3,4], Tamara Qudah [4] and Reham Tahtamouni [5]**

[1] Department of Agricultural Biotechnology and Genetic Engineering, Faculty of Agriculture Technology, Al-Ahliyya Amman University, Amman P.O. Box 19328, Jordan

[2] Department of Horticulture and Crop Sciences, Faculty of Agriculture, University of Jordan, Amman P.O. Box 11942, Jordan

[3] The Nanotchnology Center, The University of Jordan, Amman P.O. Box 11942, Jordan

[4] Hamdi Mango Center for Scientific Research, The University of Jordan, Amman P.O. Box 11942, Jordan

[5] Department of Applied Sciences, Princess Alia University College, Al-Balqa Applied University, Salt P.O. Box 19117, Jordan

\* Correspondence: r.shibli@ammanu.edu.jo; Tel.: +962-796862222

**Abstract:** Quinoa (*Chenopodium quinoa*) is of great economic importance and constitutes one of the model plants for salinity and drought tolerance in the Mediterranean climate. This study aimed to study the physiological responses of Q6 (a quinoa line developed by International Center for Bio-saline Agriculture (ICBA) in cooperation with the National Center for Agricultural Research (NARC), Jordan) grown under in vitro salt-stressing conditions (MS Media plus either 0, 25, 50, 75, 100, 150, or 200 mM of NaCl) and to determine the highest salt level that Q6 plantlets can tolerate. After this, different levels of chemically synthesized silver nanoparticles (Ag NPs) (25, 50, and 75 mg/L) were added to the growth MS media to examine if they can boost the mitigation powers of Q6 plantlets against the highest salt level that the Q6 plantlets could tolerate. Data showed that all tested growth parameters were negatively affected by adding NaCl to the media at all levels. Shoot length, proliferation, and fresh and dry weights declined to reach minimum values at 200 mM NaCl when compared to the other NaCl levels. Similarly, chlorophyll, protein, and ion content were negatively affected when exposed to NaCl at all levels, while proline increased significantly with increasing NaCl in the growth media. The addition of Ag NPs resulted in improving the mitigation powers of Q6 plantlets, especially when 75 mg/L Ag NPs were added, as this resulted in a significant improvement in microshoot growth under 200 mM NaCl when compared to the control. Adding 75 mg/L of Ag NPs to 200 mM NaCl improved shoot growth (3.56 cm) when compared to (1.04 cm) obtained plantlets that were grown in 200 mM NaCl alone. Further, other growth parameters were almost doubled by adding 75 mg/L of Ag NPs to 200 mM NaCl when compared to 200 mM NaCl alone. Additionally, adding Ag NPs (especially at the 75 mg/L level) to the media improved total chlorophyll, protein, and ion content while also reducing proline when compared to the control, which indicated an improvement in microshoot tolerance to salt-stressing conditions. These results indicate that adding specific concentrations of Ag NPs improves the growth performance and stress tolerance of Q6 grown under salt-stressing conditions.

**Keywords:** in vitro; physiological responses; quinoa; salt stress; silver nanoparticles

## 1. Introduction

Quinoa (*Chenopodium quinoa* Willd.) is an important crop that was domesticated by ancient Andean civilizations in South America [1]. It is an herbaceous, annual, and dicotyledonous plant, as well as a facultative halophyte [2,3]. It has a potential tolerance in drought and salt-stress conditions, which are commonly found in the Mediterranean region [4]. Further, quinoa also has a high nutritional value [5].

The growth and development of plants is frequently influenced by biotic and abiotic factors. Water deficiency and soil salinization are among the most abiotic stresses that affect agricultural crops [6,7]. The use of suitable crops with improved tolerances to abiotic stresses could be a logical option to consider in order for crops to cope with environmental conditions [8]. Therefore, in recent years, quinoa has become an important model crop for understanding plant responses to water and salinity stress [9]. This is because it is a stress-tolerant plant that can adapt to several environmental conditions. Indeed, the high salinity tolerance characteristics of quinoa have piqued researchers' interest.

Many studies have been conducted in order to investigate the tolerance mechanisms of quinoa plants at their many stages of development to different salinity levels [10–12]. For example, quinoa was reported to possess morphological features including stomatal density in addition to epidermal bladder cells that look similar to gigantic balloons. Further, they are about 10 times bigger than epidermal cells, which makes them able to remove 1000-fold more sodium compared to regular leaf cell vacuoles [13,14]. Additionally, quinoa was found to tolerate high levels of NaCl as they can strongly tolerate ROS, and keep a good K+ retention, maintain cytosolic Na+ at low levels, adjust fast tonoplast channel activity, and keep the pumping of H+ at high rates [3,15].

Plant tissue culture can provide suitable conditions to study stress in many plant species. This is because all of the factors and environment conditions can be easily controlled. Therefore, in vitro imposed salinity stress can be used to analyze *C. quinoa* growth and its physiological responses.

On the other hand, a range of techniques and tools can provide major benefits to the agricultural sector via the applications of modern biological research. Nanotechnology is associated with the studying of materials at length scales below 100 nm [16]. The effect of nanomaterials on a plant is versatile. However, it depends on the nanomaterials' type, size, and shape. The taking up, movement, and accumulation of nanomaterials by plants are determined by plant species [17]. Nanomaterials have been reported to ameliorate salt stress tolerance in plants. Investigations on NPs have shown that they help plants to overcome abiotic stress by their concentration-dependent impact on plant growth and development [18]. For example, nano zinc particles were used to reduce the negative effects of salinity in irrigation water on the cotton plant, where nano-Zn particles were able to reduce the P/Zn ratio in the leaves, which improved cotton plants' resistance to salinity [18]. Moreover, nanoparticles were reported to induce many antioxidant enzymes, including catalase (CAT), peroxidase (POD), and superoxide dismutase (SOD) [17]. Nanoparticles were also reported to induce micropropagation in quinoa plants when ZnONPs (2–10 mg/L) were added to the media [19]. Silver nanoparticles are considered as one of the most-generated nanomaterials for commercial products [20]. They have useful properties, such as their antimicrobial effects [21] and easily reduced (high electrochemical reduction) potential. Hence, determining the adequate concentrations of Ag NPs is necessary for determining their role in plant growth, physiology, and plant stress tolerance against different abiotic stresses. Therefore, interlinking the approach of biotechnology (plant tissue culture) and nanotechnology (silver nanoparticles) to study the physiological responses of a plant (quinoa) under induced stress can be considered an interesting field of research. No published study in the scientific literature regarding the effect of Ag NPs on in vitro-grown quinoa when under induced salinity stress has been found. Q6 quinoa is a breeding line that was developed in 2017 by the International Center for Bio-saline Agriculture (ICBA) in cooperation with the National Center for Agricultural Research (NARC) and was evaluated for its growth performance under Jordanian environmental conditions as part of the climate change adaptation and enhancement of a food security project. All experimental work was conducted in open-field conditions.

Our research is considered important as it is the first study that has investigated growth parameters and the physiological responses of quinoa (Q6 line) when grown in vitro under induced levels of salt, and how different concentrations of the chemically synthesized Ag NPs would affect Q6 line growth and its physiological responses. The importance of

our study is due to three points. First, the dearth of appropriate information on how Ag NPs would affect quinoa growth and its physiological responses when exposed to salinity. Second, the adoption of in vitro culture as a good approach that facilitates studying stresses, in general, as all other environmental factors are excluded Third, Q6 is a newly developed breeding line that has not been exposed to a tissue culture system before.

## 2. Materials and Methods

### 2.1. Establishment of Plant Materials

The seeds of the Q6 quinoa line were obtained from The National Agricultural Research Center (NARC) in Ba'qa, Jordan. The seeds of the Q6 quinoa line were surface sterilized. Then, sterilized seeds were transferred to solid MS media composed of full-strength Murashige and Skoog [22] (Murashige and Skoog, including vitamins, were purchased from the Duchefa Biochemiea company, Duchefa-Postbus 809, 2003 RV Haarlem, The Netherlands) and 30 g/L sucrose. After seed germination, plantlets (5.0 mm) were subcultured in full strength MS media (hormone-free) and supplemented with NaCl at different concentrations (0 (control), 25, 50, 75, 100, 150, and 200 mM) before being incubated under growth room conditions (the light was 16 h, the dark was 8 h, and the temperature was $24 \pm 1$ °C). The salt levels in our experiments were chosen based on previous research articles where responses of quinoa were monitored under salt stress induced ex vitro [18,19]. The replication of each treatment was five times with three shoots per replicate. Data were collected for growth parameters (shoot length; leaf number; shoots and roots; and fresh weights and dry weights (DW)), in addition to chlorophyl, protein, proline, and ion content after 4 weeks of culturing.

### 2.2. Synthesis, Characterization, and Preparation of Ag NPs

The method of nanoparticle synthesis was performed by the chemical reduction method established by Song et al. (2009) [23] with some modifications, which include the synthesis of nanoparticles via the chemical reduction method. Ag NPs concentrations (0.0, 25, 50, and 75 mg/L) were infiltered aseptically into one liter each of sterilized MS medium containing 200 Mm NaCl (the highest salt level that Q6 plantlets could tolerate, which was determined from the results of the previous experiment).

Then, plantlets (5.0 mm) were transferred to MS containing the Ag NPs and kept under similar growth room conditions as described earlier. Each treatment had five replications/treatment with three plantlets/replicate. Data were collected after 4 weeks of culturing for the similarly tested parameters, as described above.

Additionally, the plantlets' content of Ag in Q6 was determined in some samples using ICP-OES, which had shown that the Ag concentration was below the detection limit, which indicated that the Ag NPs concentrations used were very low, and thus translocation into shoot or root parts could not be observed.

Characterization of Ag NPs

UV-VIS spectrophotometry was used to obtain UV-VIS images of the nanoparticle solutions (Figure S1). The determination of the Ag concentration was achieved by ICP-OES, where the first concentration for Ag ions in $AgNO_3$ was 176.5 ppm; then, in the prepared solution, the concentration of Ag ions became 29.47 ppm, which means that the convergence % for Ag NPs = 83%.

Next, Ag NPs were characterized by means of a zeta sizer spectrophotometer of zeta potential, performed with the Malvern 3000 Zeta sizer equipment. The size was 20.27 nm, the polydisperse index (PDI) was 0.672, and the zeta potential was equal to $-33.8$, while transmission electron microscopy (TEM) was used to visualize the synthesized Ag NPs, which are smaller than visible light and thus cannot be seen by conventional optical microscopes.

### 2.3. Chlorophyll Content

The content of chlorophyll a (Chla), chlorophyll b (Chlb), and the total chlorophyll (Total Chl) were determined spectrophotometrically using equations described by Arnon (1949) [24]. Three samples (replicates) were taken from each treatment and extracts were prepared from each sample by adding 10 mL of 80% acetone to the fresh plant tissues (0.2 g), then mortar and pestle were used for grinding the tissues. Next, the extracts were transferred into a centrifuge tube and spun down (using Eppendorf 5810R Centrifuge, Eppendorf, Hamburg, Germany) at 10,000 revolutions per minute (rpm) at 4 °C for 15 min. The following step was transferring the supernatant into a fresh centrifuge tube and then the extracts were kept on ice. Next, the measuring of the absorbance at optical densities (OD) was carried out, i.e., OD645 and OD663 nm, which was achieved by using a BIO-RAD UV/Visible Spectrophotometer (SmartSpecTM Plus Spectrophotometer, Bio-Rad, Hercules, CA, USA). Total Chl, Chla, Chlb, and the ratio of Chla/Chlb were calculated according to the following equations:

$$\text{mg (Chla)}/\text{g (tissue)} = (12.7 \times \text{OD}_{663} - 2.63 \times \text{OD}_{645}) \times \frac{V}{1000 \times W} \tag{1}$$

$$\text{mg (Chlb)}/\text{g (tissue)} = (22.9 \times \text{OD645} - 4.68 \times \text{OD663}) \times \frac{V}{1000 \times W} \tag{2}$$

$$\text{mg (Total Chl)}/\text{g (tissue)} = (20.2 \times \text{OD}_{645} + 8.02 \times \text{OD}_{663}) \times \frac{V}{1000 \times W} \tag{3}$$

where *V* represents the final volume in ml, *W* represents the fresh weight of the sample by grams (g) for extracted tissue, and OD is the optical density at a specific wavelength.

### 2.4. Proline Determination

The proline concentration was determined on a fresh weight basis and a standard curve using three replicates each weighed at 0.5 g according to Bates et al. (1973) [25] with some changes. A BIO-RAD UV/Visible Spectrophotometer (Smart Spec TM Plus spectrophotometer, USA) was used for recording the absorbance at 520 nm and compared against a toluene solution treated as a blank. Using a standard stock solution of proline (100 µg), serial dilutions (0, 5, 10, 20, 30, 40, 50, 60, 70, 80, 90, and 100 µmole/mL) were prepared and the content of proline was calculated on a fresh weight basis and determined by using a standard curve, as follows:

The content of proline in (µmol g) per (FW) = (OD extract - blank)/Slop × Vol extract/Vol aliquot × 1/FW)

where FW is the fresh weight of the sample in g, OD520 is the optical density, and Vol is the final volume of the aliquot.

### 2.5. Protein Content

The content of total water-soluble protein was determined using fresh weight samples each 0.5 g according to Lowry et al. (1951) [26] with some modifications by spectrophotometer at a wavelength of 660 nm.

### 2.6. Ion Analysis

Ion analysis was carried out for calcium (Ca) content, phosphorous (P), potassium (K), and sodium (Na) according to Jones (1984) [27].

The obtained results were multiplied by the dilution factor that was used and expressed on a dry weight basis as ppm. Further, DW-1 was used for each sample. Nitrogen content was quantified according to the Kjeldahl method [28] with some modifications.

### 2.7. Data Analysis

The experimental design for all treatments in each previous experiment was arranged in a completely randomized design (CRD). Five replicates were used for the measured growth parameters and the dry weight, while three replicates were used for the determination of the plants' physiological responses. SAS analysis system software for Windows Version 9.2 (SAS Institute, Cary, NC, USA, 2004) [29] was used to analyze the collected data statistically. The obtained results were analyzed using an analysis of variance (ANOVA) and the standard error was calculated for each mean. Additionally, the mean separation was at the probability level of 0.05 as according to Tukey's HSD.

## 3. Results

### 3.1. Assessments of Plant Growth Parameters

Quinoa plant growth parameters were affected negatively by adding NaCl treatments to the nutrient media (Figure 1). For example, shoot length declined when NaCl was added in order to reach a minimum value of 1.04 cm compared to the 3.14 cm that was recorded in the control (NaCl-free media) (Table 1). A similar trend was found by the rest of the recorded growth parameters in response to adding NaCl (Table 1), where explants exposed to 200 mM NaCl were the most adversely affected (Table 1, Figure 1).

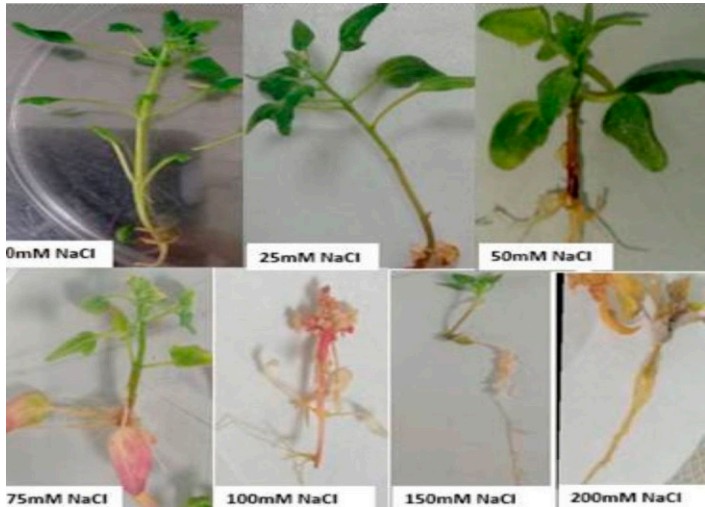

**Figure 1.** In vitro growth parameters of Q6 quinoa plantlets under the effect of (0–200) mM NaCl. Bar represents 0.5 cm.

**Table 1.** Effect of different NaCl levels on shoot length (SL), leaf number (LN), fresh weight (FW), and Dry weight (DW) for Q6 quinoa plantlets.

| Salinity Levels | Shoot Length (cm) | Leaf Number | Plant FW (mg) | Plant DW (mg) |
|---|---|---|---|---|
| 0 mM | 3.14 [a,*] | 6.90 [a,b,c] | 306.3 [a] | 17.14 [a] |
| 25 mM | 2.91 [a] | 6.10 [a,b,c] | 203.0 [a,b,c] | 13.05 [a,b,c] |
| 50 mM | 2.89 [a] | 7.50 [a,b] | 253.4 [a,b] | 15.27 [a,b] |
| 75 mM | 2.77 [a] | 8.50 [a] | 233.7 [a,b] | 14.20 [a,b,c] |
| 100 mM | 1.99 [b] | 8.00 [a] | 179.5 [b,c] | 11.81 [a,b,c] |
| 150 mM | 1.33 [c] | 4.80 [bc] | 140.7 [b,c] | 8.48 [b,c] |
| 200 mM | 1.04 [c] | 4.20 [c] | 103.00 [c] | 7.41 [c] |
| *p-values* | | | | |
| TRT | <0.0001 | <0.0001 | <0.0001 | 0.0006 |

* The data in the same column express the mean, and different letters within the columns indicate that the mean has a significant difference at the $p \leq 0.05$ probability levels.

The results (Table 2) were recorded for the Q6 plantlets that were grown in vitro under the highest salt level that they could tolerate (200 Mm). The results revealed that adding Ag NPs had improved Q6 resistance powers against salt stress as all growth parameters were enhanced after adding different Ag NPs compared to the control (Ag NP-free media). For example, adding 75 mg/L of Ag NPs to 200 mM NaCl improved shoot growth (3.56 cm) compared to (1.04 cm) obtained plantlets grown in 200 mM NaCl alone (Table 2). Values of other growth parameters were also improved in response to Ag NPs and were almost doubled by adding 75 mg/L of Ag NPs to 200 mM NaCl compared to 200 mM NaCl alone (Table 2).

**Table 2.** The growth parameters of Q6 quinoa plantlets grown under the highest NaCl level after adding different levels of Ag NPs.

| Highest NaCl Level 200 mM NaCl + Ag NP Levels (mg/L) | Shoot Length (cm) | Leaf Number | Plant FW (mg) | Plant DW (mg) |
|---|---|---|---|---|
| 200 mM + 0.0 | 1.04 [c],* | 4.20 [c] | 103.0 [b] | 7.41 [c] |
| 200 mM + 25.0 | 2.65 [b] | 5.10 [b,c] | 152.5 [b] | 10.14 [b,c] |
| 200 mM + 50.0 | 3.28 [a,b] | 6.40 [a,b] | 255.2 [a] | 13.70 [a,b] |
| 200 mM + 75.0 [1] | 3.56 [a] | 7.90 [a] | 252.8 [a] | 14.55 [a] |
| *p-values* | | | | |
| TRT | <0.0001 | <0.0001 | <0.0001 | 0.0002 |

\* The data in the same column express the mean, and different letters within the columns indicate that the mean has a significant difference at the $p \leq 0.05$ probability levels.

### 3.2. Chlorophyll Content

The content of chlorophyll a was affected in the tested lines with increasing levels of salinity (25 to 200 mM). Exposure of quinoa Q6 line to salt levels caused a gradual decline in the content of both Chla and Chlb in the Q6 leaves (Table 3). The worst results for chlorophyl were obtained in microshoots grown in (200 mM) NaCl stressing media (Table 3).

**Table 3.** Effect of different NaCl levels on chlorophyll content (fresh weight basis).

| NaCl Levels | Chla (mg/g) | Chlb (mg/g) | Total Chl (mg/g) |
|---|---|---|---|
| 0 mM | 0.9314 [a],* | 0.4591 [a] | 1.3879 [a] |
| 25 mM | 0.8278 [b] | 0.3730 [b] | 1.1986 [b] |
| 50 mM | 0.6712 [c] | 0.3245 [b] | 0.9939 [c] |
| 75 mM | 0.5825 [d] | 0.2494 [c] | 0.8304 [d] |
| 100 mM | 0.4300 [e] | 0.2345 [c,d] | 0.6632 [e] |
| 150 mM | 0.3430 [f] | 0.1728 [d] | 0.5148 [f] |
| 200 mM | 0.1832 [g] | 0.0732 [e] | 0.2559 [g] |
| *p-values* | | | |
| TRT | <0.0001 | <0.0001 | <0.0001 |

\* The data in the same column express the mean, and different letters within the columns indicate that the mean has a significant difference at $p \leq 0.05$ probability levels.

The exposure to the quinoa line to different Ag NP levels significantly enhanced Chla and Chlb synthesis in the plantlets (Table 4), where the highest chlorophyll content was recorded in the explants that were exposed to 75 mg/L Ag NPs when compared with the control treatment (Table 4).

**Table 4.** Chlorophyll content of the Q6 quinoa plantlets grown under the highest NaCl level after adding different levels of Ag NPs.

| Highest Salinity Level 200 mM NaCl + Ag NP Levels (mg/L) | Chla (mg/g) | Chlb (mg/g) | Total Chl (mg/g) |
|---|---|---|---|
| 200 Mm + 0.0 | 0.9314 [a,*] | 0.4591 [a] | 1.3879 [a] |
| 200 Mm + 25.0 | 0.8278 [b] | 0.3730 [b] | 1.1986 [b] |
| 200 Mm + 50.0 | 0.6712 [c] | 0.3245 [b] | 0.9939 [c] |
| 200 Mm + 75.0 | 0.5825 [d] | 0.2494 [c] | 0.8304 [d] |
| *p-values* | | | |
| TRT | <0.0001 | <0.0001 | <0.0001 |

* The data in the same column express the mean, and different letters within the columns indicate that the mean has a significant difference at the $p \leq 0.05$ probability levels.

### 3.3. Protein Content

The obtained data revealed that the amount of protein significantly decreased with the elevated levels of the induced NaCl in the nutrient media (Figure 2a). However, the amount of protein was significantly increased with the elevated levels of the induced silver nanoparticles in the nutrient media (Figure 2b). The protein level was two-fold higher in the stressed Q6 control plantlets than the grown plantlets in 75.0 mg/L Ag NP-supplemented media (Figure 2b).

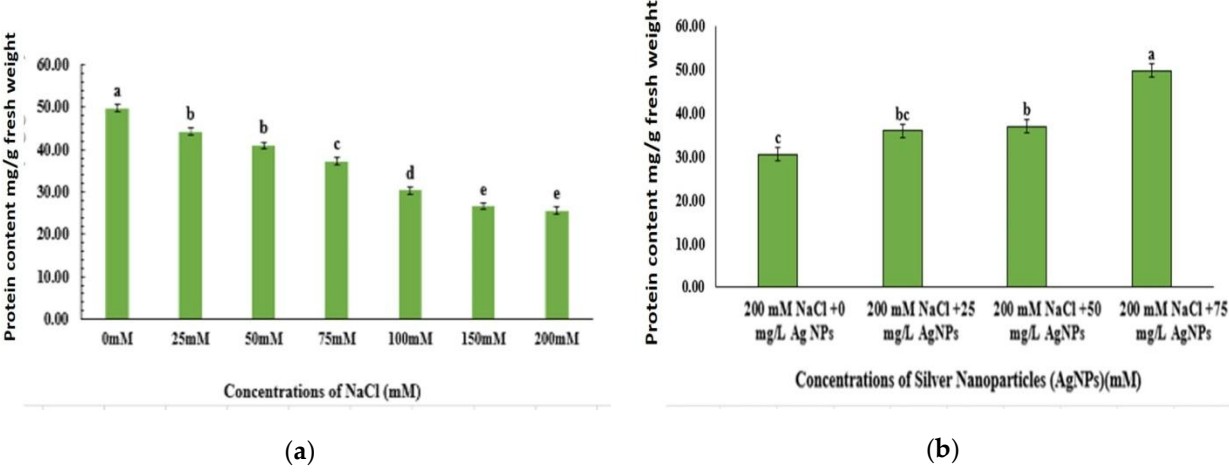

(**a**)  (**b**)

**Figure 2.** Protein content (mg/g FW) of the Q6 quinoa plantlets. (**a**) Effect of different NaCl levels on protein content (mg/g FW) of the Q6 quinoa plantlets; (**b**) effect of adding different levels of Ag NPs on the protein content of the Q6 quinoa plantlets grown under the highest NaCl level (200 mM). Data are the means ± SE of three replicates. Different letters within the columns indicate the means that have a significant difference at the $p \leq 0.05$ probability levels. Vertical bars represent the standard error at the $p \leq 0.05$ probability levels.

### 3.4. Proline Content

The salinity stress resulted in proline accumulation in the tested line (Figure 3a). The proline level was 59-fold higher in the stressed lines at salinity level 200 mM than the unstressed control. The obtained data revealed that the amount of proline significantly decreased with the elevated levels of the induced silver nanoparticles in the nutrient media (Figure 3b).

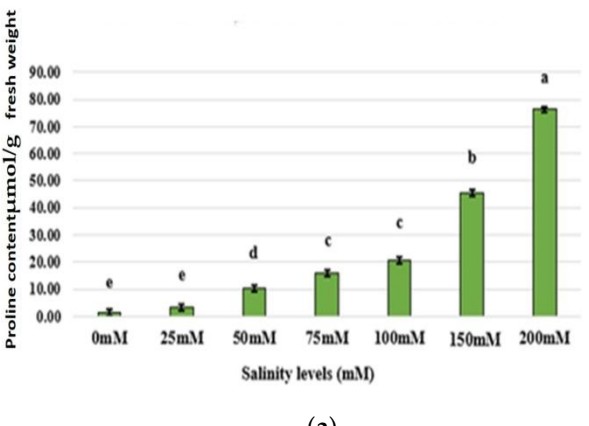
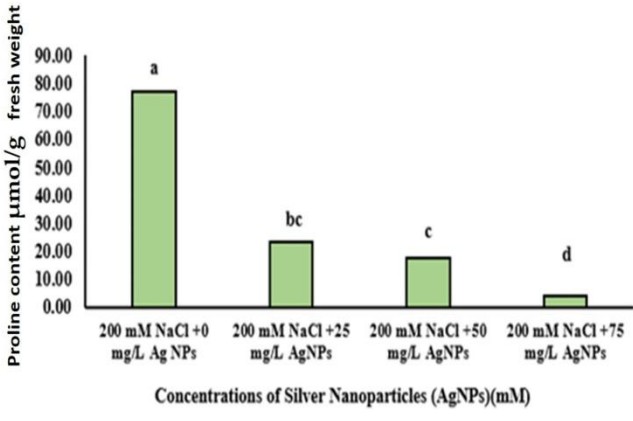

(**a**)                                                                                            (**b**)

**Figure 3.** Proline content (μmol/g FW) of the Q6 quinoa plantlets. (**a**) Effect of different NaCl levels on proline content (μmol/g FW) of Q6 quinoa plantlets; (**b**) effect of adding different levels of Ag NPs on protein content of Q6 quinoa plantlets grown under the highest NaCl level (200 mM). Data are the means ± SE of three replicates. Different letters within the columns indicate the means that have a significant difference at the $p \leq 0.05$ probability levels. Vertical bars represent the standard error at the $p \leq 0.05$ probability levels.

### 3.5. Nitrogen Content

There was a reduction in the N content in plants when subjected to different levels of NaCl in comparison to the control (Figure 4a). The highest value (7.07) was recorded for 0.0 mM NaCl when compared to the lowest value (4.10) at 200 mM NaCl (Figure 4a). Further, there was an elevation in the N content of plants that were subjected to different levels of Ag NPs when compared to the control (Figure 4b). The highest value was recorded at 75 mg/L Ag NPs. While the lowest value was at 0.0 mg/L Ag NPs (Figure 4b).

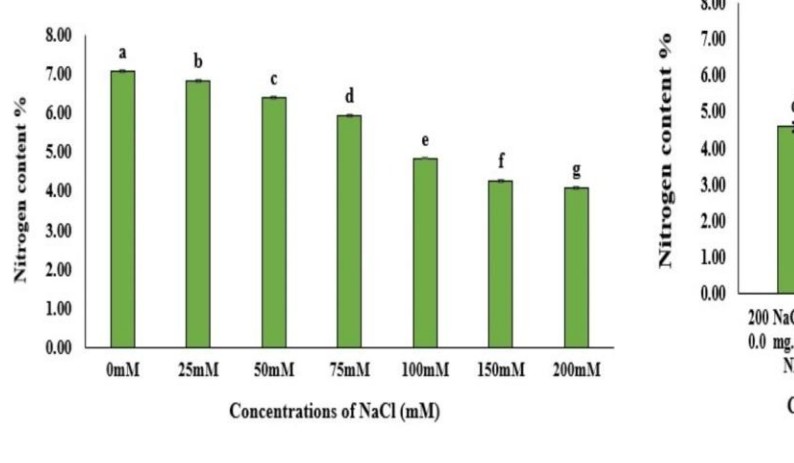
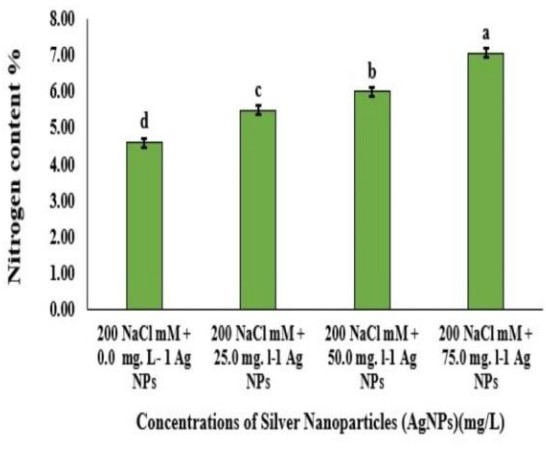

(**a**)                                                                                            (**b**)

**Figure 4.** Nitrogen content (%) of Q6 quinoa plantlets. (**a**) Effect of different NaCl levels on N content (%) of Q6 quinoa plantlets; (**b**) effect of adding different levels of Ag NPs on protein content of Q6 quinoa plantlets grown under the highest NaCl level. Data are the means ± SE of three replicates. Different letters within the columns indicate the means that have a significant difference at the $p \leq 0.05$ probability levels. Vertical bars represent the standard error at the $p \leq 0.05$ probability levels.

### 3.6. Ion Content

The data revealed that Ca content was significantly affected when exposed to NaCl at all levels. Calcium decreased as salinity level increased (Table 5), while calcium content in-

creased in the plantlets as Ag NPs was added to the most stressed media (MS supplemented with 200 mM NaCl) in contrast to the control treatment (Table 6).

**Table 5.** Effect of different NaCl levels on Ca, P, K, and Na content for Q6 quinoa plantlets (dry weight basis).

| NaCl Levels | Ca (ppm) | P (ppm) | K (ppm) | Na (ppm) |
|---|---|---|---|---|
| 0 mM (control) | 13,507.00 [a,*] | 1375.65 [a] | 492.08 [d] | 624.17 [e] |
| 25 mM | 11,655.00 [b] | 1336.82 [a,b] | 525.83 [d] | 697.08 [d,e] |
| 50 mM | 10,245.00 [c] | 1270.45 [b,c] | 555.50 [d] | 755.50 [d] |
| 75 mM | 9395.77 [d] | 1213.19 [c,d] | 584.08 [d] | 784.08 [d] |
| 100 mM | 9152.63 [e] | 1147.97 [d] | 778.75 [c] | 949.58 [c] |
| 150 mM | 8342.50 [f] | 820.14 [e] | 1445.83 [b] | 1425.58 [b] |
| 200 mM | 6792.20 [g] | 735.97 [f] | 2567.00 [a] | 2734.50 [a] |
| | | *p-values* | | |
| TRT | <0.0001 | <0.0001 | <0.0001 | <0.0001 |

\* The data in the same column express the mean, and different letters within the columns indicate that the mean has a significant difference at the $p \leq 0.05$ probability levels.

**Table 6.** The contents of Ca, P, K, and Na for Q6 quinoa plantlets grown under the highest NaCl level after adding different levels of Ag NPs.

| Highest Salinity Level 200 mM NaCl + Ag NP Levels mg/L | Ca (ppm) | P (ppm) | K (ppm) | Na (ppm) |
|---|---|---|---|---|
| 200 Mm + 0.0 | 9300.58 [d,*] | 753.69 [d] | 2537.50 [a] | 2734.50 [a] |
| 200 Mm + 25.0 | 10,615.00 [c] | 1072.68 [c] | 1350.00 [b] | 1644.17 [b] |
| 200 Mm + 50.0 | 11,951.00 [b] | 1719.39 [b] | 778.75 [c] | 949.58 [c] |
| 200 Mm + 75.0 | 13,501.00 [a] | 2670.62 [a] | 450.92 [d] | 459.42 [d] |
| | | *p-values* | | |
| TRT | <0.0001 | <0.0001 | <0.0001 | <0.0001 |

\* The data in the same column express the mean, and different letters within the columns indicate that the mean has a significant difference at the $p \leq 0.05$ probability levels.

Moreover, the obtained results indicated a significant reduction in P content with different salinity levels (Table 5). Meanwhile, the content of phosphorus increased as Ag NP levels increased (Table 6). This is especially evident at the 75.00 mg/L Ag NP level (Table 6). The contents of Na increased significantly with increasing NaCl levels in the media (Table 5), while K ions decreased in response to adding Ag NPs. Meanwhile, Na content was significantly increased with increasing NaCl level in the media (Table 5), while adding Ag NPs significantly decreased Na content compared to the control treatment (Table 6). This may indicate that Ag NPs were able to prevent Na ion absorption by the roots and thus reduced NaCl stressing effects against the Q6 plantlets, which explains the improvement in the salt tolerance powers of the plantlets.

## 4. Discussion

Salinity stress can be responsible for the unfavorable conditions that lead to a decline in agricultural production. It can alter plant metabolism, meaning that the plant tries to develop tolerance mechanisms against the stress conditions [2,3]. Quinoa, as a halophyte plant, is adapted to live in and benefit from soil containing high concentrations of salt [3]. In this study, plant growth parameters were affected at the highest level of salinity stress 200 mM, while at low levels of NaCl the quinoa plant had an acceptable growth response to the induced low levels. The reduction in growth parameters with increased salinity could indicate that the Q6 plantlets are suffering from the effect of salt toxicity [30,31]. Our results are consistent with Long (2016) [32] who reported a significant reduction in growth parameters of two quinoa genotypes in response to adding NaCl. In another

study, a significant reduction in the shoot length of quinoa plants was reported when grown in pots at a salinity level of 150 mM [14]. Additionally, Jacobsen et al. (2003) [33] reported a reduction in the biomass production of quinoa plants grown under salinity level 100–200 mM. According to Rahneshan et al. (2018) [34] in plants grown under a salt-stressed environment, nutrient uptake tends to decline as a result of low water uptake and this will reduce cell growth, thus leading to a reduction in growth and in the synthesis and accumulation of dry matter.

Since Ag NPs did not enter the plants its improvement effect was limited to its action in the nutrient medium around the plants. This effect can be due to the ability of Ag NPs to reduce ion toxicity that resulted from the high concentrations of positive sodium ions by competing and preventing sodium ions from entering the plant and thus causing a negative effect on the plant. This may refer to the high reactivity of partial positive Ag NPs when compared to the Na+ ions [35]. Therefore, Ag NPs have reduced Na transport from the roots to the shoots. Therefore, it removed the osmotic stress resulting from the increase in the concentration of sodium ions; thus, leading to an increase in plant growth [36].

These results are in complete harmony with Aghdaei et al. (2012) [37]. They revealed that the addition of Ag NPs to an in vitro-grown *Tecomella undulata* medium increased the percentage of shoot induction and the number of shoots. Similarly, compared to the control treatment, adding 50 mg/L Ag NPs was found to promote the growth and biomass of *(Brassica juncea)* seedlings [38]. Jasim et al. (2016) [39] noticed a significant increase in most of the growth parameters after adding (0.2 mg/seedling) Ag NPs to Fenugreek (*Trigonella foenum*-graecum L.) seedlings. Wahid et al. (2020) [40] stated that Ag NPs significantly reduced the adverse effect of salinity stress in wheat plants as Ag NPs were able to regulate proline accumulation, concentration of K ions, nitrogen content, and the chlorophyll content in wheat plants under salt stress. Khan et al. (2021) [41] reported that Ag NPs have the potential to reduce the induced damage from salinity stress and improve the growth of pearl millet that were grown in pots. This can be explained by the fact that Ag NPs help plants in reducing Na transport from roots to shoots in order to better tolerate salt stress [42]. In our study, Ag NPs may have helped the quinoa plants to reduce Na transport from the roots to the shoots, hence leading to an increase in plant growth.

Our results revealed that there was a significant increase in Chla, Chlb, and total Chl in Q6 quinoa at 50 and 75 mg/L Ag NPs in the salt-stressed plants, as well as in salinity stress when compared to the stressed plants alone. The same result was also obtained in another quinoa plant [43]. Chlorophyll content reduction in quinoa at higher salinity levels can be due to the degradation of the chlorophyll structure [44]. Additionally, high salt concentration tended to cause cell membrane dehydration, which would reduce $CO_2$ permeability and increased the osmotic potential that could reduce the water amount that was available to the plants [31].

It has been reported that Ag NPs significantly increased the content of Chla and total Chl in *Brassica junceae* seedlings and in *Pelargonium zonale* cultivars [38,45]. Mohamed et al. (2017) [36] obtained the same results when priming wheat plants with low concentrations of Ag NPs as this significantly increased the chlorophyll content of the stressed wheat plant when compared to the stressed plant alone. Siddiqui et al. (2014) [46] added that the application of nano-silicon dioxide ($SiO_2$ NPs), under salt stress, caused a significant increase in chlorophyll content. The improvement in chlorophyll content, in response to adding Ag NPs, could be due to a reduction in the degradation of chlorophyll and enhancement in the characteristic gas [46]. Additionally, Latef et al. (2017) [47] concluded that the content of the photosynthetic pigment in seedlings of lupine plants, primed with zinc oxide NPs (ZnO NPs), was increased under salt stress and that this is due to the presence of NPs that lowered the uptake of Na by the plants [47].

Our results showed that the application of Ag NPs caused a significant increase in the protein content of Q6 quinoa. The same results were obtained by Salama et al. (2012) [48] as they found that the addition of 60 mg/L Ag NPs to corn (*Zea mays* L.) and the common bean (*Phaseolus vulgaris* L.) grown under salt-stress conditions, caused a significant increase

in protein content. In addition, Hatami and Ghorbanpour (2014) [45] reported that the application of Ag NPs up to 60 mg/L significantly increased the protein content in the leaves of *Pelargonium zonale* cultivars.

Free proline content was described as a good index of stress that affects plants. Hence, a reduction in this parameter in the plants treated with Ag NPs was an indication of relief from stress [38]. Therefore, as proline is a stress index in the plant, the unusual decline in proline levels in our study can be considered as convincing evidence of the improved efficiency of Ag NPs when applied to the treated plantlets; further, Ag NPs may cause an enhancement of the antioxidant status of the treated plants [38]. Rossi et al. (2016) [49] obtained the same results when treating salt-stressed Brassica napus L with cerium oxide NPs as they observed a significant reduction in proline content when compared to the salt stress alone.

The data revealed that Ca content was significantly affected when exposed to NaCl at all levels. The decrease in Ca and P content in response to salinity stress was also reported by Wu et al. (2016) [50] in their study about quinoa plants. Similar results were also obtained in bean and sunflower plants when grown under salt-stressing conditions [51,52].

## 5. Conclusions

Our results indicate that adding Ag NPs to the culture media significantly improved salt tolerance in Q6 line plantlets against salt stress. However, the best tolerance performance was obtained at a silver nanoparticle level of 75 mg/L. Based on our findings, silver nanoparticles showed an amazing ability in helping Q6 plantlets to mitigate the negative effects salinity imposed under in vitro conditions. This confirms the remarkable interlinking between practices of agricultural biotechnology and nanotechnology. However, more research is needed at the cellular level for more clarification on how Ag NPs boost the mitigation powers of quinoa (Q6 line) against salinity-stressing conditions.

**Supplementary Materials:** The following supporting information can be downloaded at: https://www.mdpi.com/article/10.3390/w14193099/s1, Figure S1: Characterization of silver nanoparticles via: UV-VIS spectrophotometer and zeta sizer; Figure S2: Photos of Ag NP characterization via transmission electron microscope (TEM).

**Author Contributions:** R.S., R.M., R.A.-Z., T.Q. and R.T. conceived and designed the experiments and wrote the manuscript. R.M. and T.Q. conceived the in vitro culture work. R.M. and R.A.-Z. prepared and conducted the nanoparticle work. All authors contributed to the data analysis. All authors edited and provided a critical review of the manuscript. All authors have read and agreed to the published version of the manuscript.

**Funding:** The Deanship of Scientific research, The University of Jordan, Amman, Jordan.

**Institutional Review Board Statement:** Not applicable.

**Informed Consent Statement:** Not applicable.

**Data Availability Statement:** The datasets supporting the results of this article will be freely available upon reasonable request from RS.

**Conflicts of Interest:** The authors declare no conflict of interest.

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
