# Peer review of "Silver Nanoparticles (Ag NPs) Boost Mitigation Powers of Chenopodium Quinoa (Q6 Line) Grown under In Vitro Salt-Stressing Conditions"

_water, doi:10.3390/w14193099_

Round 1

Reviewer 1 Report

In this manuscript, authors reported that the growth and physiological responses of Quinoa (Chenopodium quinoa) plantlets under salinity stress conditions and indicated adding specific concentrations of Ag NPs would improve growth performance and stress resistance of Quinoa. Authors conducted some growth measurements included shoot length, leaves number, shoot and root fresh weights and dry weights (DW) in addition to chlorophyl, protein, proline and ion content to analyze quinoa growth and physiological responses at early stage to explore the roles of adding nanoparticles in vitro grown quinoa under induced salinity stress. Although this study may be helpful to researchers working in this field, I have some important concerns for this manuscript, the comments as follows:

1. A major problem, As we kewon, quinoa is a halophytic plant species with varieties being able to cope with salinity levels as high as those present in sea water (electrical conductivity (EC) 40 dS/m to 400 mM NaCl) [Adolf et al., 2013. Salt tolerance mechanisms in quinoa (Chenopodium quinoa Willd.) Environ. Exp. Bot., 92 (2013), pp. 43-54] and Miranda-Apodaca et al. (2018) reported up to 120 mM NaCl quinoas growth is not impaired. Please authors explain the basis for setting this concentration level (MS Media plus either 0, 25, 50, 75, 100, 150, or 200 mmol of NaC). Does a mild concentration of salt stress have a substantial effect on quinoa growth? Is the higher concentration level setting more suitable for halophytes? It is suggest that authors supplement the experimental treatment times and add some measurement of salt stress physiological indicators e.g. electron transport rate, gas exchange and chlorophyll fluorescence analysis.

2. Please authors rewrite this abstract, modify the errors (e.g. Lin 21-22) and add some important results about the effect of Ag NPs on in vitro plant growth of Quinoa.

3. In the introduction section, it is suggested that the author add some introduce about the salt tolerance mechanisms of quinoa.

4.  Please authors provide the growth photos under the treatment and control conditions.

5.  Please authors adjust the concentration units involved in this manuscript to ensure that the format isconsistent with the writing standard. (e.g. mg/L, mg/l,  mg. l-1 and mg. l-1).

6. Please authors update figures and tables to display in high definition version.

Author Response

1- Comment 1:  Answer: As we know most tissue-cultured plants are very sensitive, their shoot and rooting system are very delicate under in vitro conditions as their vascular bundles are not fully developed, so they cant withstand direct exposure to high levels of salt. So we found in the literature that quinoa was reported to grow well under ex vitro conditions when exposed to moderate salt stress (100–200 mM of NaCl), and can withstand a concentration of 400 mM of NaCl (Jacobsen, Mujica & Jensen, 2003) under ex vitro conditions. So we decided to expose our tissue cultured quinoa to salt levels within this range but in a gradual manner in order not to expose them to a salinity shock.

2- Comment 2 : Answer: the errors (e.g. Lin 21-22) were modified and some important results about the effect of Ag NPs on in vitro plant growth of Quinoa were added to the abstract as well.

3- Comment 3: Answer: In the introduction section, authors added a paragraph about the salt tolerance mechanisms of quinoa. (introduction section : LINES: 58-65. 

4- Comment 4: Answer: authors provided the growth photos under the treatment and control conditions. A Figure was provided as Figure 1

5- Comment 5: Answer: authors adjusted the concentration units involved in this manuscript to ensure that the format inconsistent with the writing standard. (they are now all in a form of mg/L)

6- Comment 6. Answer:  authors updated figures and tables to display in high definition version all figures are uploaded separately.

Reviewer 2 Report

The authors examined the effects of salt stress and silver nanoparticles on a quinoa line Q6 grown in vitro. The experimental tests revealed the mitigation of physiological responses of quinoa under salt stress by the application of silver nanoparticles. According to the Introduction, this study is novel in terms of the evaluation of the effects of silver nanoparticles on in vitro grown quinoa plants. However, the scientific value of this study is the major concern. The authors had better clarify and document what the scientific significance of this study would be. One aspect may be that the experiments were conducted in vitro, hence no microbial effects would be included in the function of nanoparticles. Another aspect may be salt stress tolerance in quinoa, and the in vitro experimental setting (system) for quinoa is useful for research on quinoa’s bioproduction or not.

Here are points to be addressed in text

(1)    What is the important point of the Q6 line? Referential line or a specific line in any aspect?

(2)    Introduction is quite poorly written comparing to Results section. Introduction needs telling readers how this study is important and valuable in science. Substantial literatures about the applications of nanoparticules (not only silver nanoparticules!) onto plant growth regulation are referred to clarify the novelty of this study.

(3)    What is the important point of silver nanoparticules? The authors have to describe the reason or say that this study focused on seeing the effects of silver nanoparticules.

(4)    Much attentions have been paid to quinoa because it could be a halophytic crop model. What examples are available on the application of nanoparticules on quinoa to improve the salt stress tolerance?

(5)    There are several errors/mistakes in English. The logical structure of the whole manuscript/sections/paragraphs and English grammars need extensive improvement/corrections. For example, single space or double space, italic or non-italic,

(6)    The format of reference list is completely correct?

Author Response

Reviewer 2

The importance of this study was better explained in the introduction section (lines: 99-115)

1- Comment 1: What is the important point of the Q6 line? Referential line or a specific line in any aspect? Answer:  The importance of Q6 line was explained in the introduction section (lines : 99-115).

2- Comment 2: Answer: Substantial works of literature about the applications of nanoparticles were added to the introduction section (lines: 76-86).

3- Comment 3:Answer:  the reason for using silver nanoparticles is explained in the introduction section (lines:83-86).

4- Comment 4: Answer: only one study was available on the application of nanoparticles on quinoa to improve micropropagation but no studies were found on the application of silver nanoparticles on quinoa to improve salt tolerance.

5- Comment 5: answer: English mistakes were corrected

6- comment 6:    The format of the reference list is completely correct. answer: also the new references were added in the correct format.

Round 2

Reviewer 1 Report

Thanks for authors' response.  I understand the plant growth characteristics  under in vitro conditions. Although I insist that the authors would add more tests especially salt treatment concentrations to verify the stability of the results and the reliability of the application. But this study is an interesting explore the effect of Ag NPs on in vitro plant growth of Quinoa that may add inspiration to follow-up research, and I agree to accept this article.